# Oxygen-binding proteins aid oxygen diffusion to enhance fitness of a yeast model of multicellularity

Whitney Wong[1], Pablo Bravo[2], Peter J. Yunker[2], William C. Ratcliff[1]*, Anthony J. Burnetti[1]*

1 School of Biological Sciences, Georgia Institute of Technology, Atlanta, Georgia, United States of America,
2 School of Physics, Georgia Institute of Technology, Atlanta, Georgia, United States of America

* ratcliff@gatech.edu (WCR); aburnetti3@gatech.edu (AJB)

**Data Availability Statement:** Numerical data and simulation code can be accessed at the GitHub repository: https://github.com/Ratcliff-Lab/oxygen-

## Abstract

Oxygen availability is a key factor in the evolution of multicellularity, as larger and more sophisticated organisms often require mechanisms allowing efficient oxygen delivery to their tissues. One such mechanism is the presence of oxygen-binding proteins, such as globins and hemerythrins, which arose in the ancestor of bilaterian animals. Despite their importance, the precise mechanisms by which oxygen-binding proteins influenced the early stages of multicellular evolution under varying environmental oxygen levels are not yet clear. We address this knowledge gap by heterologously expressing the oxygen-binding proteins myoglobin and myohemerythrin in snowflake yeast, a model system of simple, undifferentiated multicellularity. These proteins increased the depth and rate of oxygen diffusion, increasing the fitness of snowflake yeast growing aerobically. Experiments show that, paradoxically, oxygen-binding proteins confer a greater fitness benefit for larger organisms when $O_2$ is least limiting. We show via biophysical modeling that this is because facilitated diffusion is more efficient when oxygen is abundant, transporting a greater quantity of $O_2$ which can be used for metabolism. By alleviating anatomical diffusion limitations to oxygen consumption, the evolution of oxygen-binding proteins in the oxygen-rich Neoproterozoic may have been a key breakthrough enabling the evolution of increasingly large, complex multicellular metazoan lineages.

## Introduction

While molecular clock studies frequently place the origin of the metazoan clade prior to the Ediacaran period 635 million years ago [1], it is generally regarded as the period in which animal life diversified and became macroscopic [2]. Prior to this, most of the Proterozoic Eon (2.5 billion to 542 million years ago), bears relatively little trace of large complex multicellular life, with microbial mats and relatively simple multicellular algae inhabiting the oceans [3–9] alongside poorly understood microfossils that might represent relatives of the Metazoa or early sponges [10,11]. It has long been hypothesized that the dramatic rise in atmospheric

binding-proteins_paper. An archive can be found at https://zenodo.org/records/14512540.

**Funding:** W.C.R. acknowledges support from the NIH (grant no. 5R35GM138030) and NSF (CAREER Grant no. 1845363), A.J.B. acknowledges support from the Human Frontiers Science Program grant (RGY0080/2020), W.W. and W.C.R acknowledges support from the NSF Division of Environmental Biology (grant no. DEB-1845363), and P.J.Y. acknowledges support from the NIH (grant no. 1R35GM138354). The funders had no role in study design, data collection and analysis, decision to publish, or preparation of the manuscript.

**Competing interests:** The authors have declared that no competing interests exist.

oxygen levels marking the end of the Proterozoic, from approximately 1% of modern atmospheric levels to near-modern concentrations, was necessary for or triggered the evolution of animal multicellularity [12–14]. With more abundant oxygen, larger organisms would have been expected to be able to overcome diffusion limitations and achieve higher metabolic rates.

Somewhat paradoxically, recent work has shown that increasing oxygen availability can be a powerful force constraining the evolution of increased multicellular size. Oxygen is a crucial metabolic cofactor, increasing ATP returns from metabolism and allowing otherwise unfermentable carbon sources to be utilized for growth [15,16]. However, the evolution of large multicellular bodies creates a strong barrier to diffusion. This can limit the ability of interior cells to access oxygen, reducing their growth rates. The oxygenation of Earth's atmosphere thus results in counterintuitive evolutionary dynamics, constraining the evolution of macroscopic multicellular size by generating a novel and powerful trade-off between organismal size and growth rate [17,18] that did not previously exist.

To overcome these anatomical diffusion barriers and support the evolution of large body size, modern animals have evolved specialized oxygen-binding and transport proteins such as tetrameric hemoglobins, dimeric hemerythrins, and multimeric hemocyanins. These proteins are all able to bind and release oxygen cooperatively as oxygen levels change as hemoglobin does [19,20], enabling rapid loading at a respiratory organ and rapid unloading in respiring tissue. These familiar respiratory proteins are components of circulatory systems that thus enable rapid bulk transport of oxygen bound to carriers throughout the body, enabling the evolution of large, active organisms.

However, phylogenetic studies tracing the origins of animal respiratory proteins reveal that freely circulating oxygen carriers with cooperative oxygen binding in blood and hemolymph evolved well after the origin of macroscopic size, and are in fact derived from more ancient monomeric stationary globins and hemerythrins that were not transported in circulatory fluids [21–24]. These more primitive proteins were expressed in body cells and tissues, but still served to facilitate oxygen diffusion from cell to cell and through tissues by increasing the effective diffusion rate of oxygen much as modern tissue globins such as myoglobin do today [25–32]. The ancestral genes encoding these simple intracellular oxygen diffusion facilitators likely originated near or just prior to the most recent common ancestor of the Bilateria [21,23,33–35], a critical clade in animal evolution which is defined by the origin of triploblastic embryos with endoderm, ectoderm, and mesoderm germ layers which give rise to complex tissues more than 2 cells thick [36].

The selective drivers favoring the origin of facilitated diffusion via static oxygen-binding proteins remain unresolved. While it seems clear that they provide a benefit by increasing $O_2$ diffusion, no prior work has directly examined this benefit as a function of exogenous $O_2$ concentration and organism size. In this work, we examine these dynamics through synthetic biology and mathematical modeling. We heterologously express the well-studied model oxygen-binding proteins myoglobin from sperm whales (*Physeter macrocephalus*) [37] and myohemerythrin from peanut worms (*Themiste zostericola*) [38] in oxygen-limited snowflake yeast (modified *Saccharomyces cerevisiae*), a model system of diffusion-limited multicellularity capable of rapid in vitro evolution [39–42].

We expected that while enhanced oxygen diffusion would benefit snowflake yeast at all oxygen levels, the benefit would be largest at low oxygen because this environment presents the harshest limitation to the evolution of increased size in the snowflake model system [17]. However, as we examine the fitness consequences of facilitated diffusion, we find an unexpected result: oxygen-binding proteins confer the greatest advantage for large multicellular clusters not when oxygen is rare and diffusion is slow, but when it is abundant. Mathematical modeling provides insight into this counterintuitive result, showing that facilitated diffusion leads to

greater increase in oxygen flux through the surface of a cluster when oxygen levels are high compared to when they are low, alleviating anatomical limitations to the use of an abundant resource rather than compensating for low total $O_2$ availability. By directly examining the fitness consequences of proteins driving facilitated oxygen diffusion to changes in organismal size and the availability of environmental oxygen, our results provide novel context for the evolution of oxygen-binding proteins in the Metazoa.

## Results

To investigate whether heterologously expressed oxygen-binding proteins can enhance oxygen diffusion in our snowflake yeast model system, we first needed to quantify oxygen penetration depth into multicellular clusters. Prior work has shown that internal cells in snowflake yeast are strongly diffusion limited in a low-oxygen environment, with only peripheral cells able to respire actively [17].

To examine the effect of oxygen-binding proteins on the depth of oxygen diffusion, we constructed snowflake yeast strains expressing either myoglobin or myohemerythrin integrated at the HO locus (Fig 1A) and introduced the MitoLoc reporter system (preSU9-GFP + pre-COX4-mCherry) to visualize aerobic respiration. The MitoLoc system uses a constitutive GFP tag on the outer mitochondrial membrane protein TOM70 to visualize mitochondrial morphology and an mCherry tag on the inner membrane COX4 protein. This protein's rate of import from the cytoplasm to the inner membrane is dependent on mitochondrial inner membrane potential, allowing colocalization of these 2 tagged proteins to detect membrane potential indicative of high respiration rates [43].

We imaged snowflake yeast clusters expressing the oxygen-binding proteins or a wild-type control containing MitoLoc after obligately aerobic growth in Yeast Extract Peptone Glycerol media (YEP-Glycerol) under both low oxygen (10 ml cultures shaken without aeration) or supplemental oxygen levels (cultures aerated with room air bubbled through the culture [17,44]). Fractions of clusters exhibiting evidence of aerobic respiration were quantified as described in Bozdag and colleagues [17]. In low oxygen, the majority of the daily culture cycle is spent under 5% saturation (approximately 0.0125 mM). In contrast, under supplemental oxygen (see Methods for details) the majority of the daily culture cycle is spent above 50% saturation (approximately 0.125 mM) and it never drops below 32% (approximately 0.08 mM) (S1 Fig). Strains expressing oxygen-binding proteins showed significantly increased depth of oxygen diffusion under oxygen limitation to wild-type controls (Fig 1B, 21 μm mean diffusion depth for both myoglobin and myohemerythrin, versus 16 μm for the ancestor), indicating the heterologous proteins enhance penetration of oxygen to internal cells (Fig 1B and 1C). No significant difference in $O_2$ diffusion depth was observed under supplemental oxygen, however.

The fitness effects of expressing oxygen-binding proteins should depend on both the concentration of oxygen in the environment and the size of the multicellular cluster expressing these proteins. Higher environmental oxygen levels should increase the depth that oxygen penetrates, while increasing the size of the cluster should reduce the proportion of cells that are oxygenated. We examined this experimentally, by competing myohemerythrin and myoglobin-expressing strains against their isogenic GFP-tagged ancestor without oxygen-binding protein expression during aerobic growth on YEP-Glycerol. We engineered a small (approximately 10 μm radius) cluster variant as well as unicellular strains to test how size impacts the fitness advantage of oxygen-binding protein expression. Small clusters were generated through deletion of the *BUD8* landmark polarity gene, which gives rise to rounder, smaller snowflake cells, and markedly smaller groups (approximately 1.8 and approximately 5.8 times smaller in radii and volume, Figs 2A and S2). Unicells were generated via inclusion of the wild-type

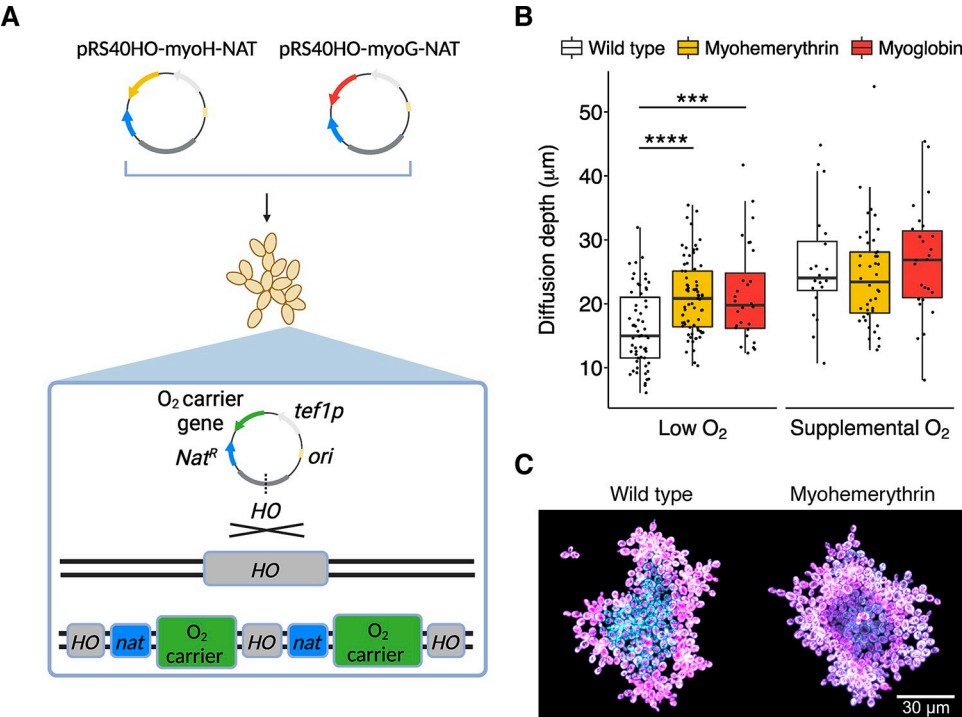

**Fig 1. Heterologous expression of oxygen-binding proteins increases O$_2$ diffusion depth in snowflake yeast.** (A) Experimental strains were constructed by inserting GFP, myohemerythrin, or myoglobin at the HO locus in the yeast chromosome. By cutting plasmids in a region of HO homology, the genes are introduced as a high-expression tandem repeat array. Created in BioRender [45]. (B) Box and whisker plot of oxygen diffusion depth using the mitochondrial MitoLoc reporter in snowflake yeast clusters expressing oxygen-binding proteins or wild-type controls under oxygen limitation. Myohemerythrin and myoglobin expression significantly increased oxygen diffusion depth in the low oxygen environment—21 μm mean diffusion depth for both myoglobin and myohemerythrin vs. 16 μm for the ancestor ($p < 0.001$, $F_{2,155} = 12.57$ one-way ANOVA, pairwise comparisons via Tukey's HSD with $\alpha = 0.05$). No significant difference was observed under supplemental oxygen ($p = 0.48$, $F_{2,85} = 0.74$ one-way ANOVA with $\alpha = 0.05$). Points indicate individual clusters. (C) Representative fluorescence micrographs of MitoLoc showing increased depth of oxygen penetration (white colocalization between GFP in cyan and RFP in magenta) in clusters expressing myoglobin (right) compared to a wild-type control (left) under low oxygen. The data underlying this figure can be found in S1 Data and at http://zenodo.org/records/14512540.

*ACE2* allele. Normal-sized clusters, small clusters, and unicells were competed under the same 2 oxygen regimes: low (cultures shaken without aeration) and supplemental (cultures aerated with room air bubbled through the liquid) (S1 Fig).

We calculated the relative fitness of each genotype against a GFP-marked control over 3 days of growth competition, with 1:100 dilutions carried out daily for multicells and 1:200 dilutions carried out daily for unicells. In small *bud8Δ* snowflake yeast, myohemerythrin and myoglobin provided no detectable fitness benefit under supplemental oxygen and a modest, marginally significant fitness advantage in low oxygen. In contrast, in normal-sized snowflake yeast clusters, both myohemerythrin and myoglobin provided a fitness benefit under all oxygen conditions. This advantage was 2.6-fold and 1.9-fold higher under supplemental oxygen relative to low oxygen conditions.

Unicellular yeast bearing myohemerythrin and myoglobin exhibited similar fitness to small *bud8Δ* snowflake yeast in low oxygen conditions, suggesting that small clusters are indeed small enough to allow oxygen to diffuse deeply into them under experimental conditions. These finesses are similar to those observed for unicells in YEP-Dextrose (see S3 Fig), an environment where fermentation dominates and respiration is of low importance, suggesting that

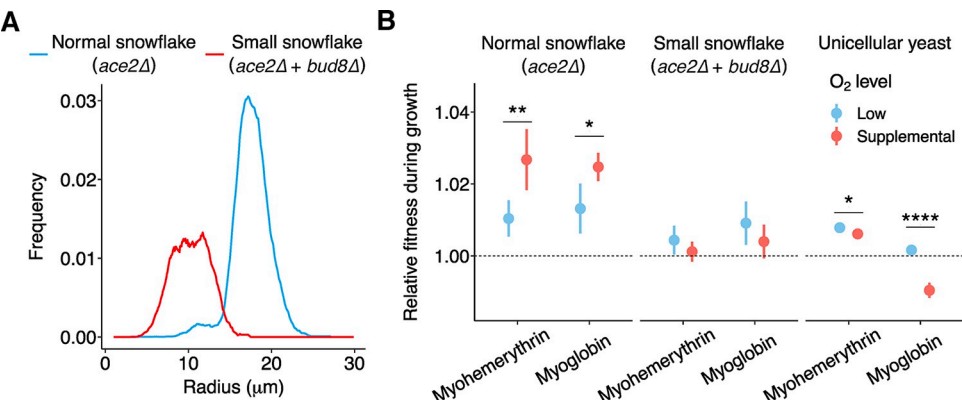

**Fig 2. Oxygen-binding proteins are most beneficial for large snowflake yeast in a high oxygen environment.** (A) Small snowflake yeast (*ace2Δ bud8Δ*), engineered by inducing a mutant that causes a random budding pattern to normal snowflake yeast, were approximately half the radius wild-type (*ace2Δ*) snowflake yeast. (B) Normal-sized snowflake yeast possessed a fitness advantage when expressing myohemerythrin (one-sample *t* tests, $p = 0.0098$, $t = 4.62$ and $p = 0.002$, $t = 7.08$, low and supplemental oxygen, respectively) and myoglobin (one-sample *t* tests, $p = 0.01$, $t = 4.24$ and $p = 0.0001$, $t = 14.06$ low and supplemental oxygen, respectively) relative to their isogenic ancestor under both low and supplemental oxygen. This advantage was 2.6-fold (two-sample *t* test, $p = 0.008$, $t = 3.72$, myohemerythrin) and 1.9-fold (two-sample *t* test, $p = 0.016$, $t = 3.25$, myoglobin) higher under supplemental oxygen relative to low oxygen conditions. The benefits of expressing myoglobin and myohemerythrin in small-sized snowflake yeast were comparatively modest. Myohemerythrin and myoglobin provided no detectable fitness benefit under supplemental oxygen (one-sample *t* tests, $p = 0.40$, $t = 0.94$ and $p = 0.13$, $t = 1.91$, respectively; Fig 2B) and a marginally significant fitness advantage in low oxygen ($p = 0.07$, $t = 2.44$ and $p = 0.03$, $t = 3.41$, respectively). In a unicellular (*ACE2*) background, myohemerythrin expression resulted in comparable fitness levels to small snowflakes with a lower slightly fitness in high oxygen than low (two-sample *t* test, $p = 0.013$, $t = 3.59$), while myoglobin showed a significant fitness defect in high oxygen compared to low oxygen (two-sample *t* test, $p = 0.000026$, $t = 10.15$). Dots represent average relative fitness with bars as one standard deviation, $n = 5$ independent competitions for each group. The data underlying this figure can be found in S1 Data and at http://zenodo.org/records/14512540.

the benefit of expression in low oxygen conditions is minimal. However, while unicellular yeast bearing myohemerythrin showed similar fitness to small clusters in supplemental oxygen conditions, myoglobin-bearing unicells exhibit a significant fitness defect of approximately 1%. We suspect that this is due to a lower maximum respiratory capacity, possibly due to abundant myoglobin proteins binding up heme or iron that would otherwise be bound for the electron transport chain. Nonetheless, this minor cost is more than made up for by the benefits of myoglobin in larger clusters (Fig 2B).

The finding that normal snowflake yeast clusters gained the greatest benefit from oxygen-binding protein expression under supplemental oxygen rather than low oxygen was surprising. We expected the impact of oxygen limitation to be most severe under low oxygen conditions like those that may have suppressed the origins of animals [14,46,47]. Why should proteins that are known for alleviating oxygen limitations be more beneficial when oxygen is plentiful? To better understand this counterintuitive result, we modeled the interplay between oxygen availability, cluster size, and myoglobin expression using reaction-diffusion equations in the Julia modeling environment (see S1 Code).

We simulated spherical yeast clusters with radii ranging from 5 to 70 μm, approximating the range of single cells to large (but not yet macroscopic) snowflake yeast that have undergone significant laboratory evolution for increased size [39–41,48,49]. Oxygen diffuses into the cluster from the surrounding environment using diffusion parameters derived from flocculating yeast [50], while being consumed aerobically. Metabolism was modeled using Monod kinetics with parameters extracted from prior literature on yeast metabolism [51–54]. Myoglobin was added to the simulation at concentrations from 0 to 0.2 mM, a concentration typical of muscle

tissue after correcting for cell packing fractions [54,55]. Oxygen was allowed to bind and unbind from myoglobin as it diffused, according to measured properties of the protein [26,56,57]. Each simulation was initiated with a snowflake yeast at a given radius, myoglobin concentration, and external oxygen concentration, and was allowed to run for 100 simulated seconds to reach equilibrium. We collected data on the final profiles of oxygen concentration across the organism, oxygen metabolism, bound and unbound myoglobin concentration, and average metabolic rates.

This computational model allowed us to predict the profile of oxygen penetration and aerobic respiration for different combinations of cluster size, environmental oxygen, and myoglobin expression. The results provide insight into how oxygen carrier proteins can enhance oxygen flux that is dependent on multicellular geometry and oxygen availability. The results of this simulation were broadly consistent with experimental observations. A typical cluster in our model (Fig 3A) exhibits a high oxygen concentration at its periphery, which rapidly falls as depth increases. This results in an outer region undergoing high rates of aerobic respiration but a hypoxic core with low or no aerobic respiration, consistent with experimental observations (Fig 1B). Notably, the addition of simulated myoglobin proteins to the model results in a significant increase in the quantity of aerobic respiration and total oxygen within the cluster—oxygen-bound myoglobin reaches equilibrium penetrating deeper below the surface of a cluster than high levels of dissolved oxygen does, contributing to aerobic respiration of inner cells. Depending on the concentration of myoglobin, this dissolved myoglobin-bound oxygen can represent a large fraction of the total oxygen flux in the system.

Further, the differential effects of oxygen-binding proteins on the growth of clusters of different sizes seen from the fitness assays (Fig 2B) is captured by this model. Experimentally, we observed that small (approximately 10 μm radius) clusters and unicells exhibited a slight fitness benefit from myoglobin expression at low oxygen levels, but no benefit at high oxygen levels. Our model indicates that at high oxygen levels, small clusters are already oxygenated to their core such that myoglobin expression does not appreciably increase the rate of respiration, while at low oxygen levels, the expression of myoglobin allows oxygen to penetrate more deeply and the average rate of aerobic respiration to increase (Fig 3B). Our model was similarly conciliant with observations that large (approximately 20 μm radius) clusters exhibited moderate fitness advantages at low oxygen levels, which rose considerably at higher oxygen levels. Our simulation indicated that large clusters are never oxygenated to their cores (which is consistent with experimental observations, Fig 1B and 1C) and so always get a fitness benefit from increased diffusion rates, with a larger total benefit arising at higher oxygen levels when facilitated diffusion has a larger impact on overall oxygen consumption.

Exploring the size and oxygen parameter space revealed an important general trend. Testing all radii from 5 to 70 μm and all oxygen levels from 0.01 to 0.25 mM (Fig 3C, simulated at a myoglobin concentration of 0.1 mM), it is apparent small clusters never gain a large benefit from expressing oxygen-binding proteins, and the little benefit they do experience is limited to low oxygen levels. As size increases, the maximum benefit obtained from facilitated oxygen diffusion and the oxygen concentration at which that benefit is maximized continually increases. It does, however, plateau at radii large enough that much of the metabolically active cluster is hypoxic. The maximum benefit increases very little above a radius of 40 μm, with the growth rate increase above that size roughly indicating the fold increase in oxygen penetration depth.

Interestingly, we find that for large clusters and high myoglobin concentrations the maximum benefit obtained from myoglobin expression is realized at intermediate oxygen concentrations of approximately 0.05 mM—roughly a fifth of saturation—with smaller but significant benefits remaining at higher oxygen levels. This is unlike the effect in small clusters, in which

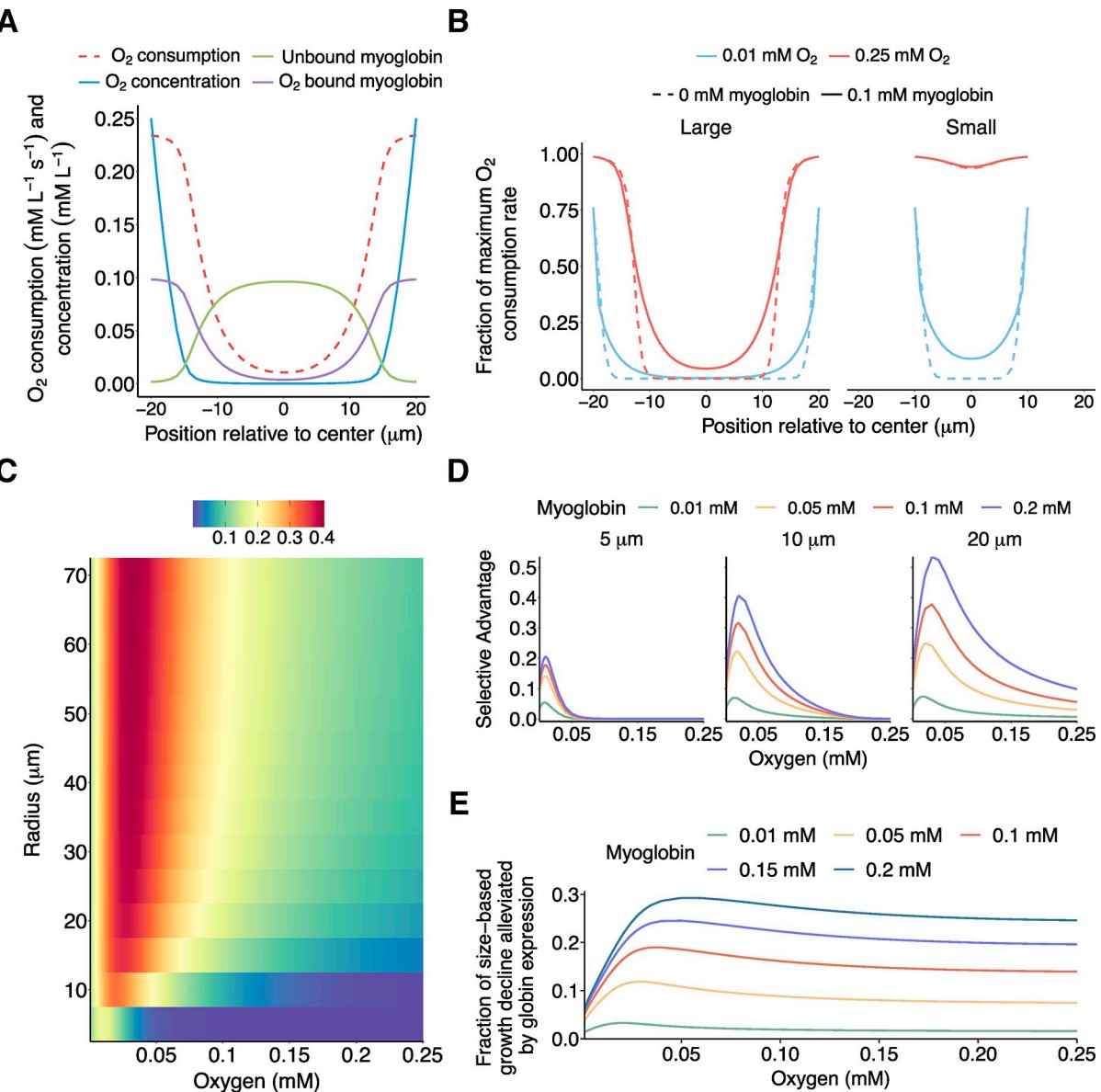

**Fig 3. Modeling the relationship between size, environmental oxygen, and the fitness effects of myoglobin.** (A) Output of a spherically symmetrical model of the coupled oxygen diffusion, myoglobin diffusion, oxygen/myoglobin binding, and aerobic respiration. Oxygen concentration falls rapidly as distance to the surface of the cluster increases, followed by oxygenated myoglobin concentration, followed by oxygen metabolism. (B) Simulated metabolic rate vs. depth of large (20 μm radius) and small (10 μm radius) clusters with and without 0.1 mM myoglobin expression. Large clusters attain larger metabolic rate increases in high oxygen environments than low; small clusters are metabolically saturated at high oxygen levels and thus attain no benefit while attaining low benefits at low oxygen levels. (C) The magnitude of myoglobin-induced metabolic rate increase, and thus myoglobin-induced selective advantage, of clusters at varied radii and oxygen levels at 0.1 mM of myoglobin expression. Selective advantage is maximized at large radii and intermediate oxygen levels. (D) Selective advantage provided by different degrees of myoglobin expression at 5 μm, 10 μm, and 20 μm radii. At small sizes myoglobin only provides benefits at low oxygen levels with severe diminishing returns to expression, as size increases myoglobin provides larger benefits at intermediate and high oxygen levels with reduced diminishing returns. (E) The fraction of growth-rate decrease caused by increasing in radius from 5 to 20 microns ameliorated by expression of different myoglobin concentrations. As the environmental oxygen level increases, the fraction of size-induced growth retardation that can be reversed by myoglobin expression rapidly increases before leveling off at intermediate values. The data underlying this figure can be found in S1 Data and at http://zenodo.org/records/14512540.

the benefits of myoglobin expression are maximized at very low oxygen levels and then decrease to zero with increasing oxygen (Fig 3D).

Examining the parameter space in more detail, we find that small clusters not only fail to obtain any benefit from myoglobin expression above low oxygen levels, but also that at these low levels the benefit that can be obtained from myoglobin expression is limited. As the quantity of myoglobin expressed increases, the benefit obtained rapidly plateaus. However, as cluster size increases, additional myoglobin begins to provide incremental increases to metabolic rate. As size and myoglobin concentrations rise, the oxygen level at which they provide their maximum benefit does as well.

Finally, we examine how much myoglobin can ameliorate the growth costs of multicellularity (that is, the growth of the group relative to that of a single cell). The fraction of growth costs ameliorated by myoglobin expression rises linearly as environmental oxygen increases from zero before leveling out at intermediate oxygen levels (circa 0.05 mM) and remains roughly constant above this level (Fig 3E). The effect is dependent upon myoglobin expression level, with rapidly diminishing returns to higher expression at low oxygen levels and slowly diminishing returns at high levels.

Taken together, this model reveals 3 important predictions for the effects of myoglobin expression on metabolic rate in respiring multicellular clusters. First, the benefits of myoglobin expression to small clusters are low and only present at all in low oxygen. Second, the metabolic benefits of myoglobin rises with increasing radius until plateauing when a large enough fraction of the total cluster volume remains hypoxic despite enhanced diffusion. Third, large clusters exhibit incremental benefits from increased myoglobin concentration with slowly diminishing returns while small clusters exhibit rapidly diminishing returns. This would explain the patterns we observed experimentally in which constitutive expression of oxygen-binding proteins exhibited large fitness benefits in large clusters and small benefits in small clusters with divergent effects of oxygen supplementation and suggests that facilitated oxygen diffusion provides its largest benefit for large diffusion-limited organisms at high oxygen levels.

To test the degree of diminishing returns of expression of oxygen-binding proteins, we put myoglobin under the control of the LexA-ER-B42 system [58] in snowflake yeast, with myoglobin expression driven by a synthetic promoter activated by a β-estradiol. This system is expected to show rapidly increasing protein expression from 0 to approximately 50 μm β-estradiol, with slower increasing expression at higher levels to approximately 200 μm [58]. Yeast were induced with 0, 10, 20, 50, and 200 μm β-estradiol and competed against GFP-bearing snowflake control clusters in low and supplemental oxygen conditions as previously described. We found that uninduced, there was no significant difference in fitness observed between low and supplemental oxygen, and that with up to 50 μm β-estradiol induction a slight cost was observed under low oxygen with significantly higher fitness observed under supplemental oxygen (Fig 4). However, further increasing induction intensity resulted in high oxygen fitness defects. We believe that further optimization of the iron import and heme production machinery of yeast, as is being explored in the bioproduction of hemoglobin and leghemoglobin [59–61], is necessary to accurately observe the dose-dependence of oxygen-binding proteins in this system at high levels of expression. Particularly, high levels of globin expression in yeast has been known to deplete heme and saturate iron import capacity at high expression levels, possibly affecting metal homeostasis and total respiratory capacity.

## Discussion

Oxygen-binding proteins appear to have originated in the common ancestor of bilaterian animals, a clade characterized by complex anatomy and thick tissues. However, the role of

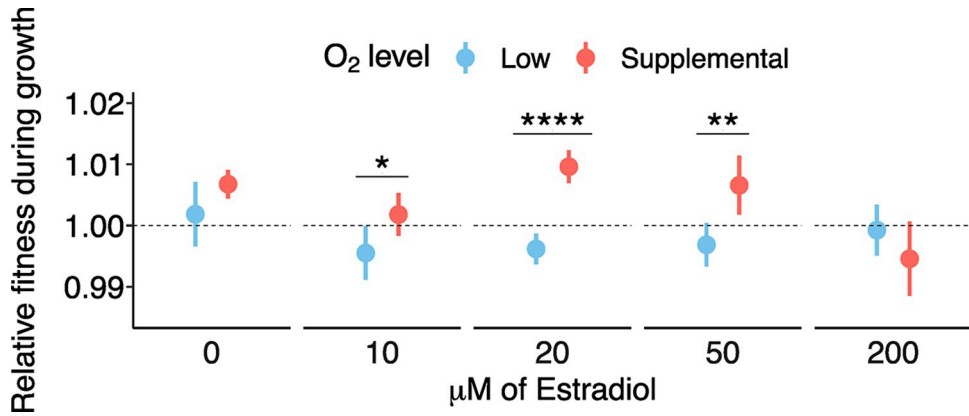

**Fig 4. Effects of varied myoglobin expression.** Experimental effect of induction of myoglobin expression at varied levels in low and high oxygen environments in snowflake yeast. Induction of myoglobin in supplemental oxygen to low (0 μm β-estradiol using shows a low level of constitutive expression) and intermediate levels (10–50 μm) shows a fitness advantage relative to low oxygen levels, with increased induction (200 μm) showing a fitness cost, while induction shows a mild fitness deficit at low oxygen (two-sample $t$ tests: 0 μm $t = -1.90$, $p = 0.11$, 10 μm $t = -2.52$, $p = 0.037$, 20 μm $t = -8.19$, $p = 0.000038$, 50 μm $t = -3.62$, $p = 0.0077$, 200 μm $t = 1.43$, $p = 0.197$). The data underlying this figure can be found in S1 Data and at http://zenodo.org/records/14512540.

oxygen-binding proteins as a key innovation facilitating the early evolution of large size has remained unclear. Much previous work has suggested that the evolution of multicellular size and complexity was limited by the availability of oxygen [12,14,46], and many modern organisms regulate myoglobin levels with increasing expression during hypoxia [62,63]. This limitation would be consistent with work on the de novo evolution of multicellularity in the snowflake yeast model system, in which $O_2$ limitation strongly impeded the evolution of larger, tougher multicellular organisms [17,64]. Facilitated oxygen diffusion, mediated by $O_2$-binding proteins, could have lifted this constraint when oxygen was low but present, allowing small multicellular organisms to overcome the constraint of low environmental oxygen.

Our results are not consistent with this hypothesis. While MitoLoc imaging showed diffusion enhancement in low-oxygen environments, we found in both competition experiments and biophysical simulations that oxygen-binding proteins provide relatively little benefit to small organisms at low oxygen levels and instead provide the greatest fitness benefit to larger organisms under higher oxygen levels. This is at first counterintuitive—why would facilitating the diffusion of oxygen not provide the greatest benefit when oxygen is least available? This can be explained by the fact that at high oxygen, the biggest oxygen limitations faced by an simple organism without a circulatory system are anatomical, arising from diffusional constraints of abundant oxygen across the organism, rather than environmental. Even when there is copious oxygen available in the environment, diffusion through rapidly metabolizing biomass nonetheless depletes it before it can reach deeply into tissues it could theoretically feed and enhanced diffusion is extremely advantageous. In contrast, when environmental oxygen is scarce, there is little metabolic benefit to be gained even by a considerable proportional increase in diffusivity because insufficient oxygen comes into contact with the surface of the organism to theoretically feed deeper tissues.

Our biophysical model illuminates the mechanism underpinning these results. The addition of globins to a system that previously lacked it (Fig 3B) causes internal oxygen gradients to become more shallow at the core of the cluster, indicating increased $O_2$ flux from the exterior, as bound myoglobin diffuses more deeply and gives up oxygen to deficient tissues. However, globin expression also leads to a steepening of the oxygen and aerobic metabolism

gradient at the surface of a cluster, as deoxygenated myoglobin diffuses outwards and takes it up. This leads to increased $O_2$ diffusion rates down this steepened concentration gradient into the organism from the environment when it is available (see S4 Fig for additional detail).

We therefore suggest an alternative hypothesis for the role and timing of the origins of oxygen-binding proteins in the Metazoa: that they may have evolved not as an adaptation to circumvent low atmospheric oxygen early in animal evolution, but instead became favored to emerge due to rising oxygen levels during the Neoproterozoic, allowing multicellular organisms to better exploit the increased metabolic potential of abundant environmental oxygen. Our hypothesis is consistent with the fact that oxygen-binding globins and hemerythrins appear to have an origin circa the common ancestor of Bilateria [21,23,33–35]. This corresponds to the Ediacaran period, after oxygen levels rose to near today's values [46,65]. An independent origin of oxygen-binding globins appears to have also occurred in land plants with nitrogen-fixing root nodules, where a similarly highly metabolically active non-photosynthetic tissue must maintain high metabolic rates despite diffusion limitations and other oxygen constraints [32,66].

The advent of dedicated oxygen-binding proteins thus appears tied to a period of rapidly increasing atmospheric oxygen levels, in organisms with thick metabolically active tissues prone to hypoxia from poor oxygen diffusion. These oxygen-binding proteins may have represented a key breakthrough, ameliorating anatomical oxygen limitations inherent to large, dense, metabolically active multicellular organisms, thereby enabling continuing increases in organismal size and complexity. Our results support the hypothesis that this ancestral emergence of simple intracellular oxygen-binding proteins aided early multicellular lineages in overcoming size constraints. Rather than evolving as a response to environmental constraints to oxygen diffusion in $O_2$-poor environments, oxygen-binding proteins likely emerged because of the oxygen abundance of the Neoproterozoic, allowing early animals to overcome anatomical diffusion limitations and exploit the increased metabolic potential of this newly abundant resource. By integrating theory and experiments, this work develops a new perspective on how innovations in oxygen utilization facilitated increases in multicellular scale and complexity, highlighting the importance of global environmental change in shaping the pattern and process of major evolutionary transitions.

## Methods

### Strain construction

All *S. cerevisiae* strains were constructed starting from the homozygous diploid Y55 derivative Y55HD previously used by the Ratcliff laboratory [41] and strain GOB8 bearing an *ACE2* deletion (*ace2Δ*::KANMX/*ace2Δ*::KANMX) also previously described [17,40,41,49]. We synthesized yeast codon-optimized reading frames coding for peanut worm *(Themiste zostericola)* myohemerythrin [27], sperm whale *(Physeter macrocephalus)* [26,37] myoglobin, and GFP via Thermo Fisher gene synthesis and ligated them into a custom expression vector under the control of a *TEF1* promoter with *NATMX6* resistance cassette and a region of *HO* homology for chromosomal insertion. Plasmids were inserted into the chromosome by cutting within the HO homology using AfeI or BsaAI. This method of integration generates a tandem repeat array of the expression vector integrated into the chromosome at the *HO* locus bracketed by repeats of the *HO* homology region. Due to myoglobin requiring heme to form a functional protein, we enhanced heme production in myoglobin expression lines and GFP control lines by replacement of the *HEM3* gene promoter (the rate-limiting enzyme in heme production in *S. cerevisiae*) by the *TEF1* promoter and a *HYGMX6* selectable marker via standard PCR-based methods. This generated "normal-sized" snowflake strains (see strain list). "Small-sized" yeast

for fitness assays were created by first deleting both copies of the *BUD8* gene in snowflake yeast using standard PCR-based methods using the *HYGMX6* marker in addition to previously described manipulations, generating 4 additional strains (see strain list). All genotypes were constructed by standard yeast husbandry techniques involving sporulation and mating of modified strains.

MitoLoc-bearing strains used to visualize and quantify oxygen diffusion were created by inserting a single copy of the *preSU9-GFP* + *preCOX4-mCherry* construct from the MitoLoc plasmid (Addgene #58980) into the same HO-integrating expression vector. This plasmid was cut for genomic insertion at the *HO* locus with AfeI or BsaAI to generate 2 heterozygous Mito-Loc and oxygen-binding protein expressing snowflake strains (see strain list).

## Plasmids used

pYM25: *GFP* for protein fusion and an *hphNT2* resistance cassette [67].

pWR86: *TEF1* promoter and *hphNT2* for gene overexpression.

pWR78: Expression of *P. macrocephalus myoglobin* [37] under a *TEF1* promoter, *NATMX6* resistance, *HO* homology for chromosomal insertion by cutting with BsaAI or AfeI.

pWR79: Expression of *T. zostericola myohemerythrin* [38], under a *TEF1* promoter, *NATMX6* resistance, *HO* homology for chromosomal insertion by cutting with BsaAI or AfeI.

pWR162: Expression of *GFP* under a *TEF1* promoter, *NATMX6* resistance, *HO* homology for chromosomal insertion by cutting with BsaAI or AfeI.

pWR236: Expression of the *LexA-ER-AD* β-estradiol-inducible transcriptional activator under an *ACT1* promoter, *HO* homology for chromosomal insertion by cutting with BsaAI or AfeI.

pWR244: Expression of *myoglobin* under a β-estradiol-inducible promoter with 4 *LexA* boxes (*4xLexApr*), *HO* homology for chromosomal insertion by cutting with BsaAI or AfeI.

## Oligonucleotides used

Bud8_S2
TACCCAATATCCTCTTTCTACTTGAGAATTTTTTCGATTCTACATGAAGTatcgatgaattcgagctcg

Bud8_R
GACAGAACAGTTTTTTATTTTTTATCCTATTTGATGAATGATACAGTTTCgacatggaggcccagaatac

Hem3_oe_F
AAAGCGAAGAAAATCTAGATAAATTTGTAGTTGGTAAATACACACGTACTccagatctgtttagcttgcc

HEM3_oe_R
ACCGCCAATTTCGATTTTCTCCCACCAATATGTAGAGTTTCAGGGCCCATcttagattagattgctatgctttc

## Strains used

yAB623: Y55HD background, *ace2△:KanMX4/ace2△:KanMX4, ho:MyoH:NatMX6/ho*:*MyoH:NatMX6*

yAB626: Y55HD background, *ace2△:KanMX4/ace2△:KanMX4, tef1pr-HEM3:HphNT2/tef1pr-HEM3:HphNT2, ho:MyoG:NatMX6/ho:MyoG:NatMX6*

yAB632: Y55HD background, *ace2△:KanMX4/ace2△: KanMX4, tef1pr-HEM3:HphNT2/tef1pr-HEM3:HphNT2, ho*:*GFP:NatMX6/ho:GFP:NatMX6*

yAB635: Y55HD background, *ace2△:KanMX4/ace2△:KanMX4, ho:GFP:NatMX6/ho:GFP: NatMX6*

yAB714: Y55HD background, *ace2△:Kanmx4/ace2△:Kanmx4, bud8△:HphNT2/bud8△: HphNT2, tef1pr-HEM3: HphNT2/tef1pr-HEM3: HphNT2, ho:MyoG:NatMX6/ho:MyoG: NatMX6*

yAB718: Y55HD background, *ace2△:Kanmx4/ace2△:Kanmx4, bud8△: HphNT2/bud8△: HphNT2, tef1pr-HEM3:HphNT2/tef1pr-HEM3:HphNT2, ho:GFP2:NatMX6/ho:GFP2:NatMX6*

yAB723: Y55HD background, *ace2△:KanMX4/ace2△: KanMX4, bud8△:HphNT2/bud8△: HphNT2, ho:MyoH:NatMX6/ho:MyoH:NatMX6*

yAB727: Y55HD background, *ace2△: KanMX4/ace2△: KanMX4, bud8△: HphNT2/ bud8△: HphNT2, ho:GFP2:NatMX6/ho:GFP2:NatMX6*

yAB708: Y55HD background, *ace2△:KANmx4/ace2△:KANmx4, HEM3/tef1pr-HEM3: HphNT2, ho:MyoG:NatMX6/ho:MitoLoc:NatMX6*

yAB710: Y55HD background, *ace2△:KanMX4/ace2△:KanMX4, ho:MyoH:NatMX6/ho: MitoLoc:NatMX6*

yAB1019: *tef1pr-HEM3:HphNT2/tef1pr-HEM3:HphNT2, ho:GFP:NatMX6/ho:GFP:NatMX6*

yAB1020: *ho:GFP:NatMX6/ho:GFP:NatMX6*

yAB1021: *tef1pr-HEM3:HphNT2/tef1pr-HEM3:HphNT2, ho:MyoG:NatMX6/ho:MyoG: NatMX6*

yAB1023: *ho:MyoH:NatMX6/ho:MyoH:NatMX6*

yAB1043: Y55HD background, *ace2△:KanMX4/ace2△:KanMX4, tef1pr-HEM3:HphNT2/ tef1pr-HEM3:HphNT2, ho:4xLexApr-MyoG:NatMX6/ho:act1pr-LexA-ER-B42:NatMX6*

## Imaging MitoLoc

To quantify oxygen diffusion depth, the engineered MitoLoc strains were grown in YEP-Glycerol (1% yeast extract, 2% peptone, 2.5% glycerol) in order to ensure their energy metabolism was fully dependent upon respiration, and 10 ml cultures were grown in a low oxygen environment shaking at 250 RPM at 30˚C, and supplemental oxygen cultures were aerated by bubbling humidified laboratory air through the growth media at 30˚C using 14-gauge needles as described in Bozdag and colleagues [17] with protocols for the use of multiplexed chemostat arrays [44]. After 6 to 7 h of growth post-inoculation, each strain was imaged using a Nikon Eclipse Ti inverted microscope under TRITC and FITC channels. The diffusion depth of oxygen for each strain was determined as described in Bozdag and colleagues [17]. Clusters were flattened under a coverslip before imaging. Flattened clusters were segmented automatically in imageJ, and inner anaerobic regions not exhibiting red and green colocalization were manually segmented within them. Areas were measured, and a summary statistic of oxygen diffusion depth was calculated by subtracting the radius of a circle with the area of the anaerobic region from the radius of a circle with the area of the full cluster.

## Fitness assays

All strains were competed against GFP-bearing control lines with the same *ACE2*, *BUD8*, and *HEM3* genotypes. All yeast were grown in 10 ml of appropriate culture media (YEP-Glycerol or YEP-Dextrose) at 30˚C shaken at 250 RPM. Low oxygen cultures were stirred in a shaking incubator without aeration, or aerated as described above from Bozdag and colleagues with protocols for the use of multiplexed chemostat arrays [44]. Monocultures of all necessary strains were grown for 24 h to stationary phase in YEP-Glycerol or YEP-Dextrose as needed for 1 day, and then 100 microliters reinoculated into another tube for a second day before experiments began. On experiment day 0, equal volumes of GFP-expressing control yeast and

experimental yeast were mixed to form the starting population. Five replicate populations were inoculated from each mixture. A total of 100 microliters of this mixture was inoculated into 10 ml of fresh media at the start of each competition in the case of multicellular constitutive expression experiments, while 50 microliters was inoculated for unicellular competitions and β-estradiol induction competitions. The populations were passaged for a total of 3 days, inoculating the same volume of culture into fresh media every 24 h. Upon initial mixing of monocultures on day 0, and after day 3 of growth, 25 μl of multicellular cultures were diluted into 1 ml of water on 12-well plates and allowed to settle. Tiled bright-field and FITC channel images were obtained via the Nikon Eclipse Ti inverted microscope, until 1,000 to 3,000 clusters were imaged. Clusters were automatically segmented via ImageJ scripts, segmentation errors manually corrected, and clusters under 250 square microns automatically excluded. Each cluster was automatically classified as GFP or non-GFP based on fluorescence level using an R script, thresholded based on the FITC channel intensity with minimal cluster density. Unicellular cultures were instead counted via flow cytometry and analyzed via FloMax 3.0 (Partec, Göttingen, Germany).

After measuring the final proportions of GFP and non-GFP clusters or unicells for each population, we calculated the relative fitness of each strain relative to its GFP control by finding the ratio of their Malthusian parameters [68]. Specifically, given an initial population fraction of $p_0$, a population fraction on passage $n$ of $p_n$, and a dilution factor of $f_d$, the final relative fitness $W$ of that population fraction is expressed as follows:

$$W = \frac{\ln(f_d{}^n \cdot p_n / p_0)}{\ln(f_d{}^n \cdot (1 - p_n)/(1 - p_0))} .$$

## Mathematical modeling of the dynamics of oxygen, globin, and size

The generic 1-dimensional diffusion equation in Cartesian coordinates of the concentration of generic substance $f$ with respect to time $t$ and position $x$ is as follows, where variable $D_f$ corresponds to the diffusion constant of substance $f$, giving us Eq 1:

$$\frac{df}{dt} = D_f \cdot \frac{d^2 f}{dx^2} \tag{1}$$

To transform the diffusion equation into 3 dimensional symmetrical spherical coordinates, with $x$ now corresponding to a radius from the center of a sphere, we use the following Eq 2:

$$\frac{df}{dt} = \frac{1}{x^2} \cdot D_f \cdot \frac{d}{dx}\left(x^2 \cdot \frac{df}{dx}\right) \tag{2}$$

This can be rearranged into the final model's version of the diffusion equation in polar coordinates, resulting in Eq 3:

$$\frac{df}{dt} = D_f \cdot \left(\frac{d^2 f}{dx^2} + \frac{2}{x} \cdot \frac{df}{dx}\right) \tag{3}$$

The generic variable $f$ can be replaced by the variable $o$ for free oxygen concentration, $m_u$ for unbound myoglobin concentration, and $m_b$ for bound myoglobin concentration, all in units of mMol/L. The generic diffusion constant $D_f$ can be replaced by the diffusion constant $D_o$ for the diffusion of oxygen or $D_m$ for the diffusion constant of myoglobin (equal for both bound and unbound to oxygen), both in units of $\mu m^2 s^{-1}$.

Free oxygen consumption was modeled via the Monod equation [69]. This requires a maximum rate of oxygen consumption $o_{max}$ in mMol/L/s (noting that the volume given is the volume of modeled multicellular organism including empty space as well as cell volume), and a

Monod constant $k_u$ for the concentration of oxygen at which the consumption rate is at half-maximum in mMol/L. Change in oxygen concentration caused by oxygen consumption is given by Eq 4:

$$\frac{do}{dt} = -o_{max} \cdot \frac{o}{k_u + o} \tag{4}$$

Myoglobin's interaction with free oxygen is described via an association constant $k_f$ and a dissociation constant $k_r$. The variable $k_f$ is measured in mM$^{-1}$s$^{-1}$, while the variable $k_r$ is measured in s$^{-1}$, both determined experimentally. The first derivative of free oxygen with time due to association and dissociation with myoglobin is given by Eq 5:

$$\frac{do}{dt} = -k_f \cdot o \cdot m_u + k_r \cdot m_b \tag{5}$$

The first derivative of unbound myoglobin with time due to association and dissociation with oxygen is given by Eq 6:

$$\frac{dm_u}{dt} = -k_f \cdot o \cdot m_u + k_r \cdot m_b \tag{6}$$

The first derivative of bound myoglobin with time due to association and dissociation with oxygen is given by Eq 7:

$$\frac{dm_b}{dt} = k_f \cdot o \cdot m_u - k_r \cdot m_b \tag{7}$$

To define the final model differential equation for oxygen concentration with respect to time, we add the equations for the first derivatives of oxygen with respect to time as a result of diffusion, association/dissociation, and oxygen consumption, resulting in Eq 8:

$$\frac{do}{dt} = D_o \cdot \left( \frac{d^2 o}{dx^2} + \frac{2}{x} \cdot \frac{do}{dx} \right) - k_f \cdot o \cdot m_u + k_r \cdot m_b - o_{max} \cdot \frac{o}{k_u + o} \tag{8}$$

To define the final model differential equation for unbound myoglobin concentration with respect to time, we add the equations for the first derivatives of unbound myoglobin with respect to time as a result of diffusion and association/dissociation, resulting in Eq 9:

$$\frac{dm_u}{dt} = D_m \cdot \left( \frac{d^2 m_u}{dx^2} + \frac{2}{x} \cdot \frac{dm_u}{dx} \right) - k_f \cdot o \cdot m_u + k_r \cdot m_b \tag{9}$$

To define the final model differential equation for bound myoglobin concentration with respect to time, we add the equations for the first derivatives of bound myoglobin with respect to time as a result of diffusion and association/dissociation, resulting in Eq 10:

$$\frac{dm_b}{dt} = D_m \cdot \left( \frac{d^2 m_b}{dx^2} + \frac{2}{x} \cdot \frac{dm_b}{dx} \right) + k_f \cdot o \cdot m_u - k_r \cdot m_b \tag{10}$$

Eqs 8 through 10 were implemented in the Julia modeling environment, with a fixed boundary condition of constant external oxygen concentration.

## Parameter values used

While total myoglobin concentration $m_t$ (equal to the sum of $m_b$ and $m_u$), external oxygen $o_{ext}$, and radius $r$ were variables reinitialized with every model run; most variables in this model

were constants and were obtained from a survey of the scientific literature. Myoglobin association constant $k_f$ and dissociation constant $k_r$ were calculated for heart myoglobin by Endeward [26] based on numbers from Antonini [57]. Association constant $k_t$ was taken to be 15,400 mM$^{-1}$s$^{-1}$, and dissociation constant $k_r$ was taken to be 60 s$^{-1}$.

While the diffusion constant of oxygen in pure water is measured as over 2,000 μm$^2$s$^{-1}$ [70], diffusion is highly limited in the crowded and heterogenous interior of a cell with many cell walls to cross. Vicente and colleagues [50] has directly measured the diffusion constant of dissolved oxygen in dense yeast flocs, with variations in experimental technique resulting in measurements between 4.9 and 29.3 μm$^2$s$^{-1}$. We used the average of their measurements, 17.1 μm$^2$s$^{-1}$. Similarly, the diffusion constant of myoglobin is taken to be 16 μm$^2$s$^{-1}$, as measured in the crowded environment of muscle tissue by Papadopoulus and colleagues [56].

The Monod constant $k_u$ of yeast oxygen consumption is provided by model fitting of experimental data by Sonnlietner and Käppelis [51] as 0.1 mg/L, equivalent to 0.003125 mMol/L. Yeast maximal metabolic rates are seldom measured in terms of the required units of flux per unit cellular volume, however. It is more frequently measured in oxygen flux per unit dry mass. The maximum value of oxygen consumption per unit dry mass of yeast was taken to be 8 mMol/g/hr, also from Sonnlietner and Käppelis [51]. To convert oxygen consumption rate per unit dry mass of yeast into oxygen consumption rate per unit volume of a multicellular cluster, a dry mass fraction of total mass, a density, and packing fraction is additionally required.

The water mass fraction of yeast is given as 0.604 by Illmer and colleagues [52], resulting in a dry mass fraction of 0.396. The density of live Y55 yeast is given as 1.1126 g/ml by Baldwin and Kubitschek, 1984 [53]. Multiplying these together with the maximum oxygen consumption rate per unit dry mass gives a maximum oxygen consumption rate per unit cellular volume of 0.791 mMol/L/s. However, the snowflake yeast being modeled are not 100% tightly packed cellular volume, instead having a packing fraction resulting from significant free space between cells. According to the dissertation of Dahaj [54], unevolved snowflake yeast such as those measured in this work has a packing fraction of approximately 0.3. Multiplying this packing fraction by the maximal oxygen consumption rate per unit cellular volume gives a final $o_{max}$ value of 0.237 mMol/L/s.

## Numerical simulations

Eqs 8 to 10 were implemented as a set of coupled differential equations in Julia version 1.8.2. The Julia model is provided (S1 Code) in the form of a Jupyter notebook, project file, and manifest file. In all models, the internal oxygen concentration $o$ at all points was initialized at 1e-5 mM, and the bound myoglobin concentration $m_b$ was initialized in equilibrium with this. The initial unbound myoglobin concentration $m_u$ at all points was initialized to be equal to the total remaining myoglobin concentration required to reach $m_t$. Total myoglobin concentration $m_t$, the external dissolved oxygen boundary condition $o_{ext}$ (also in mMol/L), and the radius of the cluster $r$ (in μm) were all variables provided with the initialization of a given model run. Models were run for 100 s of simulated time to reach equilibrium. Models were discretized with a radius step size of 1 μm, and the total oxygen consumption rate according to the Monod equation (Eq 4) summed across all discrete radii, to produce a total oxygen consumption rate. This total rate was divided by the volume of the cluster to determine the average metabolic flux per unit volume of a cluster of a given radius in a given environment. The variable $o_{ext}$ was allowed to vary from a low of 0.001 to a high of 0.25 mMol/L, the approximate maximum solubility of oxygen in water at 30°C [71,72]. The radius $r$ was allowed to vary from 5 μm to 100 μm. The total myoglobin $o_{ext}$ was allowed to vary from 0 to 0.2 mM/L, a value

**Table 1. Parameter table.**

| Variable | Value |
| --- | --- |
| $D_o$ ($\mu m^2 s^{-1}$) | 17.1 |
| $D_m$ ($\mu m^2 s^{-1}$) | 16 |
| $k_f$ ($mM^{-1} s^{-1}$) | 15,400 |
| $k_r$ ($s^{-1}$) | 60 |
| $o_{max}$ (mMol/L/s) | 0.237 |
| $K_u$ (mMol/L) | 0.003125 |

similar to that which is observed in typical animal heart muscle tissue [55] multiplied by the packing fraction of 0.3 [54].

Parameter table (see Table 1)

# Supporting information

**S1 Data. Collated data represented in all main and supplemental figures.**
(XLSX)

**S1 Code. Code necessary for running coupled diffusion model of myoglobin and dissolved oxygen.** Includes a Jupyter notebook, manifest file, and project file.
(ZIP)

**S1 Fig. Oxygen levels encountered during ordinary and supplemental oxygen growth cycles.** Without supplemental oxygen, the majority of a growth cycle is spent at oxygen levels under <5% saturation (<0.0125 mM). With supplemental oxygen, the majority of a growth cycle is spent at greater than 50% oxygen saturation (>0.125 mM) and average oxygen levels do not drop below 32% saturation (0.08 mM). Data taken from Bozdag and colleagues [17]. Five replicate measurements of oxygen profiles taken with fiber-optic optodes were averaged to produce each oxygen profile. The data underlying this figure can be found in S1 Data and at http://zenodo.org/records/14512540.
(TIFF)

**S2 Fig. "Normal" snowflake yeast (left) and a genetically engineered "Small" strain (*bud8Δ*, right).** Deletion of the *BUD8* gene, which encodes a protein that plays a role in pole selection for budding, results in smaller snowflake yeast clusters. Deletion results in daughter cells back-budding towards mother cells thus fracturing clusters and creating a smaller cluster phenotype.
(TIFF)

**S3 Fig. Fitness of unicellular yeast under different growth conditions.** The fitness of unicellular strains when grown in YEP-Dextrose when oxygen was not required. Y55 base strain unicells and myohemerythrin-bearing unicells showed a similar fitness advantage of approximately 0.7% over GFP-bearing competitors, indicating a low cost to myohemerythrin, while the cost of myoglobin expression is roughly equivalent to GFP (one-sample $t$ tests, $p = 0.000018$ $t = 23.89$, $p = 0.16$ $t = 1.75$, and $p = 0.0000052$ $t = 32.7$ for myohemerythrin, myoglobin, and Y55, respectively). This cost of myoglobin is significantly lower than was observed when grown in YEP-Glycerol under low oxygen conditions, while the cost of myohemerythrin is similar. The data underlying this figure can be found in S1 Data and at http://zenodo.org/records/14512540.
(TIFF)

**S4 Fig. The effect of 0.1 mM globin expression on modeled oxygen gradients within a metabolizing cluster at high (0.25 mM) oxygen concentration.** (A) The expression of globin steepens the oxygen gradient at the surface of a cluster, while shallowing the oxygen gradient deeper within a cluster. As the Monod constant of yeast oxygen consumption corresponds to a low oxygen concentration of approximately $3*10^{-3}$ mM, the oxygen consumption rate (B) of the deeper portions of the cluster with low oxygen is affected much more strongly than the consumption rate of the shallower portions with high oxygen. Thus, the increase in oxygen consumption deep within the cluster outweighs the slight decrease in oxygen consumption close to the surface of the cluster. The data underlying this figure can be found in S1 Data and at http://zenodo.org/records/14512540.
(TIFF)

# Acknowledgments

We would like to thank members of the Ratcliff and Yunker laboratories for constructive feedback on this paper.

# Author Contributions

**Conceptualization:** Pablo Bravo, William C. Ratcliff, Anthony J. Burnetti.

**Data curation:** Whitney Wong, Anthony J. Burnetti.

**Funding acquisition:** William C. Ratcliff.

**Investigation:** Whitney Wong, Anthony J. Burnetti.

**Methodology:** Whitney Wong, Peter J. Yunker, Anthony J. Burnetti.

**Project administration:** William C. Ratcliff.

**Resources:** Peter J. Yunker, William C. Ratcliff.

**Software:** Pablo Bravo, Peter J. Yunker.

**Supervision:** Peter J. Yunker, William C. Ratcliff, Anthony J. Burnetti.

**Visualization:** Whitney Wong, Pablo Bravo, William C. Ratcliff.

**Writing – original draft:** Whitney Wong, William C. Ratcliff, Anthony J. Burnetti.

**Writing – review & editing:** Whitney Wong, Pablo Bravo, Peter J. Yunker, William C. Ratcliff, Anthony J. Burnetti.

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
