## [Editor Report · Decision Letter 0]

15 Feb 2024

Dear Dr Burnetti, 

Thank you for submitting your manuscript entitled "Examining the role of oxygen-binding proteins on the early evolution of multicellularity" for consideration as a Research Article by PLOS Biology.

Your manuscript has now been evaluated by the PLOS Biology editorial staff, and I'm writing to let you know that we would like to send your submission out for external peer review. I should say that we were not able to secure an assessment from an Academic Editor in a timely manner, but have decided nevertheless to sent it out for peer review.

IMPORTANT: While you submitted your paper as a regular Research Article, we think that it would be better considered as a Short Report. Your paper is already very concise, so no re-formatting is required. Please select "Short Reports" as the article type when you upload your additional metadata (see next paragraph).

Once your full submission is complete, your paper will undergo a series of checks in preparation for peer review. After your manuscript has passed the checks it will be sent out for review. To provide the metadata for your submission, please Login to Editorial Manager (https://www.editorialmanager.com/pbiology) within two working days, i.e. by Feb 19 2024 11:59PM.

Kind regards,

Roli Roberts

Roland Roberts, PhD

Senior Editor

PLOS Biology

rroberts@plos.org

---

## [Decision Letter · Decision Letter 1]

15 Apr 2024

Dear Dr Burnetti,

Thank you for your patience while your manuscript "Examining the role of oxygen-binding proteins on the early evolution of multicellularity" was peer-reviewed at PLOS Biology. It has now been evaluated by the PLOS Biology editors, an Academic Editor with relevant expertise, and by three independent reviewers. 

You'll see that reviewer #1 is very positive about the study, but asks if you could re-run the model using myohemerythrin, wants them to discuss the role played by morphology, and has a number of textual and presentational requests. Reviewer #2 says that s/he can’t judge the modelling, but is overall very positive and most of their requests are textual changes to enhance the clarity. Reviewer #3 wants you to check the effects of the O2-binding proteins on the growth dynamics of both snowflake and ancestral yeast, wants more robust measurement of O2 diffusion depth, asks you to try a non-bud8 snowflake strain, and wonders if you could use an inducible promoter to titrate the O2-binding proteins more subtly.

IMPORTANT: I discussed these comments with the Academic Editor, including how much further work we should expect for a Short Report. I've pasted a summary of their advice at the foot of this email, and you'll see that they considered several of the experimental requests to be optional, but that two of them would significantly strengthen the paper.

In light of the reviews, which you will find at the end of this email, we would like to invite you to revise the work to thoroughly address the reviewers' reports.

Given the extent of revision needed, we cannot make a decision about publication until we have seen the revised manuscript and your response to the reviewers' comments. Your revised manuscript is likely to be sent for further evaluation by all or a subset of the reviewers.

**IMPORTANT - SUBMITTING YOUR REVISION**

*Re-submission Checklist*

*Published Peer Review*

*PLOS Data Policy*

*Blot and Gel Data Policy*

Sincerely,

Roli Roberts

Roland Roberts, PhD

Senior Editor

PLOS Biology

rroberts@plos.org

REVIEWERS' COMMENTS:

Reviewer #1: 

[IMPORTANT: See attachment for the formatted version!]

Summary:

The report by Whitney Wong et al. focuses on a relevant theme of evolutionary developmental biology: How oxygen permeability limited/facilitated the evolution of large and likely complex multicellular organisms. The collected experimental and theoretical evidence presented in the report led to a fascinating hypothesis on oxygen-binding proteins' role during this evolutionary transition. Authors found that the exogenous expression of oxygen-binding proteins gave a higher fitness advantage to large multicellular groups at high O2 levels than at low O2 levels. This was not the case for small multicellular groups since the expression of oxygen-binding proteins did not provide any significant fitness advantage under any circumstances. Exploring more in-depth the potential dynamics of oxygen diffusion in the presence of heme-proteins using biophysical modelling, the authors argue that oxygen-binding protein played a crucial role in the evolution of complex 

multicellularity during the Ediacaran period once oxygen levels had significantly increased.

All the following comments and points are meant to increase the quality of the already exciting work I had a chance to read. I enjoyed reading and studying for this report, and hopefully, I have accomplished my task by providing a helpful review.

Specific comments:

The synthetic approach used in this work is fascinating and engaging, although explaining how the specific oxygen-binding proteins were chosen could be helpful. Still, I appreciated using two different proteins from different organisms instead of one only. On a side note, I found it interesting that myoglobin and myohemerythrin showed similar results in the experimental work.

• Minor point: Assuming that myohemerythrin has different kinetic properties than myoglobin, would it be possible to run the model for myoH? Given the experimental evidence, it should align with the trends of myoG despite potential biochemical differences. Would you expect differences? If yes, could these inform us on the evolution of oxygen-binding proteins?

As pointed out in the manuscript, oxygen diffusion can be limited by size increase. However, in the context of the report’s question, ‘size’ can be applied to spherical multicellular groups (the valid approach used in the biophysical model), where size directly reflects cell layers or thickness (e.g., Lines 312-314). This distinction made me observe the microscopy images shown in Fig 1C closely and return to those previously published in Bozdag et al. 2021. The wild type suggests a strong association between morphology and permeability, such that those cells that are close (i.e., below only one cell) or directly exposed to the environment can show/or not show signs of respiration. In Bozdag et al. 2021, this is the case for intermediate O2 conditions. How can morphological evolution potentially limit the adverse effects of poor oxygenation on multicellular growth? Can limited accessibility to oxygen constrain the evolution of specific morphs, such that growing as a spherical body would have been negatively selected in a low-oxygen environment? This seems to be a relevant aspect that

might need to be expanded if space allows.

Relevant to Fig1:

• Minor point: It would be helpful to show representative micrographs of MitoLoc yeast expressing Myoglobin and all genotypes under high O2 levels. If space is a limitation, a supplementary panel could be provided.

• Minor point: The figure caption of Fig. 1A needs to include information reported in the main text. I don’t see the point of having such a level of detail in a main figure without making it directly understandable within the figure caption. The figure could be simplified, and the additional information could be moved to a supplementary figure.

• Minor point: Currently, the figure caption indicates that Fig 1B reports mean with SD. However, I interpreted these to be boxplots reporting IQR, medians, and whiskers (?). Moreover, I couldn’t find the number of biological replicates. Are individual points considered biological replicates? This would need to be clarified.

• Minor point: It could be convenient to consider making images accessible to colourblind readers. The reported micrographs will convey little if no message at all. I list the common potential colorblind-friendly two-colour alternatives:

Green/Magenta, Yellow/Blue, and Red/Cyan. Fig 3 has a similar issue but is less limiting than in Fig 1C. I am sure you already know, but in ImageJ, you can recolour individual channels easily (Image>Color>Channel Tools).

Fig. 2B reports the exciting contrast relative to the fitness gains between low and supplemental O2 in normal and small snowflakes. However, I found it difficult to understand why this result was unexpected, even considering the cited references (Lines 156-158).

Normal heme-expressing snowflake yeast gains a fitness advantage under low O2 levels relative to the wild type (Lines 138-142). Moreover, the significant difference in permeability shown in Figs 1B and C is detected under low O2 in contrast to the supplemental O2 level condition, which shows no significant difference.

Main-minor point: I consider the O2 levels the primer limiting factor for its diffusion, even if myoH or myoG are abundant. This means that O2 needs to be available to these proteins to observe any effect. In low oxygen conditions, O2 is likely enough only to sustain the physiological activity of cells sitting at the multicellular group periphery.

The results obtained from the model go in this direction (Fig. 3B). The presence/absence of heme proteins is relevant if enough oxygen can be transported. This is also congruent with the observation that, in small snowflakes, there is (yet a non-significant) increase in fitness at low O2 for myoglobin.

Importantly, my point is still congruent with the authors’ conclusion, which I find valid and interesting; I am only raising the question of whether the results from the biophysical model could already be expected by considering the experimental data and previous literature. My specific knowledge of the evolution of multicellularity and the constraints imposed by oxygen availability is likely less than that of the authors; thus, I could be missing some additional information to properly follow the above-mentioned transition.

Additional minor points:

Fig. 3 – Expressing ‘cluster radius from the centre’ as negative values seems odd (i.e., reporting distances using negative values). Since all graphs are symmetric to 0, they could be halved (as in Fig S3).

Line 157 – Intermediate is likely low in this context.

Methods – It could be relevant to cite or describe how distances were measured in Fig. 1B.

Reviewer #2:

[identified himself as Tanai Cardona]

Wong et al. report experiments suggesting that oxygen-binding proteins only became evolutionary advantageous once oxygen was an abundant resource, and in multicellular organisms that were complex enough for diffusion of oxygen alone to be insufficient as a delivery mechanism to inner cells. The authors have backed up the experimental work with mathematical modelling. Wong et al. state that the results challenge some of the current thinking on the topic, which, according to the authors, is based on intuitive notions.

I find the data and discussion in alignment with the scope of Short Reports at PLOS Biology. The data is compelling and clearly presented, and it supports the conclusions well. So, I don't have any major criticism.

I should disclose that I'm unable to evaluate the methodology regarding mathematical modelling.

My commentary below is aimed at enhancing clarity.

These are presented in sequential order:

Line 26-30. Boring billion. I wish to encourage you to be a little more precise in these statements. A lot of evolutionary important events occurred within this period, including the origin of eukaryotes and their early diversification into many lineages. It sounds as if the world was only inhabited by prokaryotes and algae from 2.5 to 0.5 billion years ago.

Line 27. Boring billion refers more specifically to the range between 1.8 to 0.8 billion years ago, not the entire Proterozoic. It should be noted that the Proterozoic is actually two billions! :)

I wonder if you can replace some of the references 1 to 7 with something more recent that captures better the state of knowledge. Have a look at this reference, just one example: https://www.nature.com/articles/s41586-023-06170-w.

Line 32. Reference [8] is not displaying correctly in the reference list. Also, what do you mean with "linked"? It could be read as a causal relationship. Try to use more precise language if possible.

Line 73-74. It might be a good idea to explain why this is "unexpected". For example, has the opposite been suggested by others previously? If so, can you provide some reference for this? Or was your original hypothesis the opposite to what you found when you set out to do these experiments?

Consider your introduction in light of your discussion section. You might want to explicitly state at the beginning what had been hypothesised before, or what the expected/intuitive hypothesis was. Then, explain how your work disproves it.

Lines 94-95. I'm not sure how broadly used this method is but consider rephrasing this for extra clarity. For example, say that, according to the reference, COX4 is imported into mitochondria proportional to the membrane potential, which is proportional to the rates of respiration (if I understood correctly). It just feels a bit vague. 

Line 98-100. I wonder if you can use more precise language here. For example, "after 7 hours the concentration of oxygen in the culture was below 5% saturation..." It would be ideal if you could mention the concentrations of dissolved oxygen in concentration units, rather than as a percentage of saturation.

Line 274. "Linked", how? In which way were these linked?

Line 275, "This has been…" it's not clear what "this" refers to.

Lines 271-280. I just get a feeling that this works better in the introduction.

Line 303. "In response to", not precise enough. Perhaps evolved "as an adaptation to"...

Lines 330-332. It isn't clear the rationale for choosing myohemerythrin and myoglobin from the mentioned organisms. Can you provide a brief statement on this? Are these standards in the field?

Line 351. Please rephrase for clarity. "Previously described" Where? Could you be more precise?

Line 398. What did you use to bubble air?

Line 400-401. It isn't clear how you do this. Use more precise language. How do you determine depth from an area measurement, when these cluster of cells are highly irregular? Might be a good idea to provide a visualisation.

Line 404-408. Is the number of cells measured by OD? Do you expect the number of cells to be approximately equal per unit of volume? Why don't you just simply say the exact volumes of culture grown at the start of the experiment, and the volumes used for each step of the experiment? 

Line 411/Line 414-415. How did you measure the final proportions for each population? How many cells do you count? Do you count with a software or manually? How many replicates?

Line 412. "finding the ratio of Malthusian parameters" you provide three references for this. Can you be more precise as to where exactly the complete information needed can be found? Think about a new student wanting to reproduce these experiments and measurements.

Line 669. "vanishingly" can you be more precise? I would personally equate "vanishingly low" with few parts per million.

Lines 671-672. Fig. S1 Do you mean that this data is taking from ref. 11? It isn't clear. Please clarify this in the materials and methods. Also what oxygen-detection method was used to produce the traces in Fig. S1. 

Reviewer #3:

[identifies himself as Omaya Dudin]

In this work, the authors use the experimentally evolved snowflake yeast to assess the importance of oxygen-binding proteins in bypassing the diffusion barrier within a multicellular colony. This is an interesting and appealing system, be it a genetically tractable system that can be manipulated in various ways to ask simple question about the transition to multicellularity. In that sense, the experiment done here is elegantly thought and snowflakes are very suitable to tackle this question. The unintuitive result obtained, in which bigger-sized colonies have a greater fitness benefit at high O2 compared to small cells is somewhat supported by the data, and the use of a computational model seems to support these results. Overall, it's an interesting and exciting result, and would be suitable for a short report in PloS Biology. However, some concerns below need to be addressed to strengthen some conclusions and make this a better study.

1) growth dynamics of budding yeast cells with plasmids with myoglobin and myohemerythrin. Many experiments assume that these strains grow at a comparable rate as the WT, although there is no data for this. I think ensuring that cells grow at same speed is critical for measurements of O2 depth but also fitness. I would recommend assessing the growth dynamics in the snowflakes Ace2- as well as in the single-cell ancestor Ace2+. Ensuring that these strains grow at similar speed is critical for interpreting following data.

2) The measurements of oxygen diffusion depth in principle should be alright done using microscopy. However, to strengthen this I recommend a better explanation of the measurement's method, maybe with a drawing and comparative kymographs. Another point that I think is critical, is accounting for overall colony size. Small size differences between colonies may allow for increased variability. Maybe measuring depth per colony and normalizing it per colony diameter for each colony would better help to obtain a more robust measurement.

3) The small snowflake colonies are generated through a bud8 mutation. Although there is no known link between bud8 and metabolism, we can also imagine that for some indirect reason, bud8 interacts negatively with myoglobin etc. It would strengthen the conclusions if there was another mutation, maybe wee1 or for3

4) The predictions from the computational model concerning the linear increase with increase in amounts of myoglobin should be easily testable experimentally. Although I understand this is a short report, it could be envisaged to test this using inducible promoters or promoters of different strengths. 

Small things:

- mutant names should be italic

- Line 15 to 8 is copy pasted from abstract, a variation is better for reader.

- You need a ref for Line 27th

- Statistical test description can be added to figure legends rather than within Figure callouts. It breaks the reader flow

- Fig S2: the scale bars are not of similar sizes.

GUIDANCE FROM THE ACADEMIC EDITOR: [edited]

 a) The AE said: I find the following statement (and repeated several times in the text) a bit annoying: “Experiments show that, paradoxically, oxygen-binding proteins confer a greater fitness benefit for larger organisms under high, not low, O2 conditions.” To me, this is completely expected, for the same reasons that their model suggests. I suggest deleting "paradoxical" to make it less confusing.

b) "re-run the model using myohemerythrin" (rev #1) - AE thinks this is optional.

c) The AE said "With regard to reviewer 3, I also wondered about the effects of the O2-binding proteins on the growth dynamics of both snowflake and ancestral yeast." It would be good to assess this.

d) "robust measurement of O2 diffusion depth" (rev #3) - AE thinks this is optional.

e) "try a non-bud8 snowflake strain" (rev #3) - AE thinks this is optional.

f) "use an inducible promoter to titrate the O2-binding proteins more subtly" (rev #3) - the AE says "this would be a nice ending to the paper. Their model predicted dosage dependence, and if that is demonstrated experimentally, it will make the paper stronger."

---

## [Decision Letter · Decision Letter 2]

8 Dec 2024

Dear Dr Burnetti,

Thank you for the submission of your revised Short Reports "Examining the role of oxygen-binding proteins on the early evolution of multicellularity" for publication in PLOS Biology. On behalf of my colleagues and the Academic Editor, Wenying Shou, I'm pleased to say that we can in principle accept your manuscript for publication, provided you address any remaining formatting and reporting issues. These will be detailed in an email you should receive within 2-3 business days from our colleagues in the journal operations team; no action is required from you until then. Please note that we will not be able to formally accept your manuscript and schedule it for publication until you have completed any requested changes.

IMPORTANT: You'll see that the three reviewers are fully satisfied with you revisions and have no further requests. However, I will be asking my colleagues to include the follow very minor requests alongside their own:

a) Please could you change the Title to include an active verb and to make it more explicit? We suggest "Oxygen-binding proteins aid oxygen diffusion to enhance fitness of multicellular snowflake yeast" or "Oxygen-binding proteins aid oxygen diffusion to enhance fitness of a yeast model of multicellularity"

b) Many thanks for your comprehensive provision of data and code. Please cite the location of the data clearly in all relevant main and supplementary Figure legends, e.g. “The data underlying this Figure can be found in S1 Data” or “The data underlying this Figure can be found in https://zenodo.org/records/XXXXXXXX

c) I note that you already have an associated GitHub deposition (https://github.com/TonyBurnetti/Oxygen-BindingProteins); because Github depositions can be readily changed or deleted, please make a permanent DOI’d copy (e.g. in Zenodo) and provide this URL in addition.

Sincerely, 

Roli Roberts

Senior Editor

PLOS Biology

rroberts@plos.org

REVIEWERS' COMMENTS:

Reviewer #1:

[identifies himself as Marco La Fortezza]

The authors addressed all previously raised points with great detail. The effort in answering all reviewers' comments significantly strengthened an already excellent paper. Therefore, I support the publication of this work, which brings new and relevant insight into the molecular constraints/facilitations caused by oxygen diffusion for the evolutionary transition to multicellularity.

Reviewer #2:

[identifies himself as Tanai Cardona]

Thank you for the detailed response to my recommendations. All my points were addressed satisfactorily. I have no further comments.

Reviewer #3:

I must thank the authors for their very careful review of our previous comments.

Two experiments made this new version very strong; The assessment of growth across the species and the inducible promoter experiment. Both are strengthening the conclusions and making this story a very cool one and solid one.

Congrats to the author.

No further comments on my side.